# Serum Retinol but Not 25(OH)D Status Is Associated With Serum Hepcidin Levels in Older Mexican Adults

**DOI:** 10.3390/nu11050988

**Published:** 2019-04-30

**Authors:** Vanessa De la Cruz-Góngora, Aarón Salinas-Rodríguez, Salvador Villalpando, Mario Flores-Aldana

**Affiliations:** 1Center for Evaluation and Survey Research, National Institute of Public Health, Cuernavaca 62100, Mexico; asalinas@insp.mx; 2Nutrition and Health Research Center, National Institute of Public Health, Cuernavaca 62100, Mexico; svillalp@insp.mx (S.V.); mario.flores@insp.mx (M.F.-A.)

**Keywords:** hepcidin, vitamin A, vitamin D, older adults

## Abstract

(1) Background: Elevated hepcidin levels have been linked to anemia of inflammation (AI). Retinol deficiency has shown to upregulate hepcidin expression in animals, while conflicting evidence links VD status with hepcidin concentration in humans. The purpose of the study is to explore if VA and VD status are associated with hepcidin concentrations in older Mexican adults (OA). (2) Methods: A cross-sectional study was conducted in summer 2015, using serum samples from 783 fasting OA ages 60 and above residents from Campeche and Yucatán. VA deficiency (VAD) was defined as serum retinol concentration <20 μg/dL and VD deficiency (VDD) as 25(OH)D <50 nmol/L. The log-hepcidin was the outcome variable expressed as continuous and tertiles of its distribution. Linear and ordinal regression models were used. (3) Results: VAD was present in 3.4% and VDD in 9.5% of OA. Log-retinol was inversely associated with log-hepcidin (coeff.: −0.15, 95%CI: −0.2, −0.09). VAD status shown a higher probability than non-VAD for higher hepcidin tertiles (OR = 2.15, 95%CI: 1.24, 3.74). VDD states was not associated with hepcidin in the linear (coeff.: 0.16, 95%CI: −0.02, 0.34) nor the ordinal model (OR = 0.74, 95%CI: 0.42, 1.28). (4) Conclusions: VAD, but not VDD, status was inversely associated with hepcidin concentrations in OA.

## 1. Introduction

Hepcidin is the main hormone that regulates iron homeostasis and has been considered the main mediator of anemia of inflammation (AI) or anemia of chronic disease [1]. Hepcidin causes a block in iron recycling through the hepcidin-ferroportin axis, thus diminishing iron export from macrophages during iron overload or infection despite sufficient iron stores [2]. In older adults (OA), inflammation, metaflammation, immunosenescence, and frailty are all conditions that conferred a dysregulation of the immune system, promoting a low and chronic proinflammatory profile [3,4,5]. Although the pathological mechanism is still poorly understood, interleukin (IL)-6 seems to induce hepcidin expression under inflammatory conditions, therefore serving as a risk factor for AI development in OA [1,2,6].

Vitamins A (VA) and D (VD) play a crucial role on the immune response, since both exert their effects on target cells by binding to nuclear-hormone receptors. In interaction with either RAR or RXR nuclear receptors, both vitamins may directly regulate gene expression [7]. A vitamin D receptor element (VDRE) in the promoter region of the *Hepcidin Antimicrobial Peptide* (*HAMP*) gene has been identified, suggesting that VD may downregulate hepcidin expression independently from immunomodulation of pro-inflammatory cytokines [8]. VD deficiency (VDD) has been associated with anemia in healthy and diseased populations [7,9,10]. Additionally, some studies have explored the effect of VD supplementation on hepcidin levels in adults, most of which showed a reduction in hepcidin levels after the improvement in VD status [8,11,12].

VA deficiency (VAD) has been found to induce upregulation of *HAMP* expression in rodents [13,14]. Although no data currently exist to link VA status to circulating hepcidin levels in the adult population, associations between VA status and the effect of VA supplementation on anemia has been well documented in infants and women [15]. It is likely that iron mobilization, in addition to the different actions of VA on erythropoiesis, may be one of the mechanisms through which VA exerts its role to maintain iron homeostasis [16].

VDD is highly prevalent in older adults across different populations, while VAD exists principally in developing countries [10,17,18]. Previously, we found that hepcidin levels in OA were higher in those with AI and CKD anemia in comparison to non-anemics, and that VA status, but not VD, showed an association with AI [19]. Identifying those immunomodulatory metabolites that might be associated with higher hepcidin concentrations is relevant for the prevention of anemia with an inflammatory component. The aim of the study was to explore if serum retinol levels and 25(OH)D levels are both associated with hepcidin concentrations in OA, in a manner independent of anemia status.

## 2. Materials and Methods

From July through September 2015, we recruited 829 OA (ages ≥60) for a cross-sectional study to study the causes of anemia in OA [19]. Participants were recruited from four localities in the southern region of Mexico, including Champotón, Campeche, Mérida, and Valladolid, and interviewed at their homes.

For the sampling procedure, we used a stratified multistage cluster sample design. If in the household two OA were living together, only one was randomly selected to participate in the study. Selection criteria: all ambulatory OA were invited to participate in the study. Exclusion criteria applied to OA with any leg amputation, dementia (or complete dependence of their caregiver), or were using supplementary oxygen, or if for some other condition they were immobilized or resting full-time in bed.

Of total study subjects, 803 had available serum and hematological parameters. Information on sociodemographic characteristics, chronic comorbidities, anthropometry, diet, nutritional status, and education was gathered by trained research assistants during home visits. The study was approved by the Ethics, Biosecurity and Research committee at the Instituto Nacional de Salud Pública (National Institute of Public Health). All participants gave their informed oral and written consent.

### 2.1. Biochemical Analysis

Fasting venous blood samples were drawn and centrifuged in situ between 6:00 and 9:00 am, during previously scheduled appointments. The serum was then separated and stored in coded cryovials, and preserved in liquid nitrogen until delivery to a central laboratory in Cuernavaca, Mexico, where it was stored at −70 °C.

VD (serum 25(OH)D) (nmol/L), C reactive protein (CRP mg/dL), Homocystein (umol/L), B12 (pg/mL) and Ferritin were measured by immunochemiluminescence method using commercials kits (Abbott Lab, Illinois, IL, USA) in an ArchitectCI8200 equipment (Abbott Diagnostics, Wiesbaden, Germany). Serum retinol was determined in an HPLC HP1110 LCDAD (Agilent Technology Waldbronn, Germany), using NovaPack C18 4um 3.9 × 150 mm with a flux of 1.5 mL/min of the mobile phase methanol, after extraction with 99% ethanol. Hepcidin-25 was measured through a quantitative immunoassay technique using a commercial immunoassay (MyBioSource ELISA kit, San Diego, CA, USA) with a detection range between 4.69 ng/mL–300 ng/mL. Capillary hemoglobin was measured using a portable photometer (Hemocue, Angelholm, Sweden). Serum iron, was measured with an atomic absorption spectrophotometer (Agilent Technology, Waldbronn, Germany). Soluble transferrin receptor (sTfR) was measured using a commercial immunoassay (Quantikine IVD sTFR ELISA kit; R and D Systems Inc., Minneapolis, MN, USA) using recombinant human sTFR as standards. Alpha glycoprotein 1-acid (AGP), erythropoietin (EPO) and IL-6 concentrations were measured through immunoassay ELISA, using commercials kits (R and D Systems Inc., Minneapolis, MN, USA). Creatinine (mg/dL) was measured through a colorimetric method with an ArchitectCI8200 analyzer (Abbott Diagnostics, Wiesbaden, Germany).

Biochemical analyses were performed at the Centro Médico Nacional Siglo XXI and at the Nutrition Laboratory at the National Institute of Public Health in Mexico. 

### 2.2. Definition of Variables

VD status was classified as deficient if serum 25(OH)D was <50 nmol/L [20] and VA was considered deficient if serum retinol <20 μg/dL [21]. Anemia was classified according to WHO criteria as Hb <13 g/dL in men and <12 g/dL in women [22]. Iron deficiency (ID) was defined as sTfR >28 nmol/L or serum ferritin concentration <15 ng/mL, adjusting for inflammation [23]. High ferritin status was defined as ferritin ≥350 ng/mL, and low serum iron as <60 μg/dL. Vitamin B12 deficiency (B12D) was defined if Fedosov’s equation fell below −0.5 SD, accounting for serum homocysteine levels and folate [24]. Glomerular filtration rate (GFR) was estimated from serum creatinine with the use of the CKD-EPI. Chronic kidney disease (CKD) was defined if estimated GFR <60 mL/min/1.73 m^2^ [25] or a previously-diagnosed kidney disease. IL-6 values were categorized as >10 pg/mL [26]. Categories of inflammation were considered using AGP and CRP combination according Turnham: reference, incubation, early convalescence, and late convalescence [23].

Body mass index (BMI) was calculated as weight in kilograms divided by the squared height in meters, and study participants were classified as normal (18–24.9 kg/m^2^) or overweight/obese (≥25 kg/m^2^). Presence of chronic diseases (hypertension, diabetes, dyslipidemia, myocardial infarction, angina pectoris, heart disease, cirrhosis, arthritis, stroke, chronic lung disease, osteoporosis, and cancer) were obtained by self-report if previously diagnosed by a physician. Use of total medication was registered and classified as either non-steroid anti-inflammatory drugs (NSAID) or steroid anti-inflammatory drugs (SAID). Ethnicity was defined based on which indigenous language was spoken in the household. An asset index was considered using a principal component analysis, the first component gave a total variance of 33%. This was divided into tertiles with the uppermost tertile indicating the highest socioeconomic status (SES).

The phenotype of frailty was the proposed by Fried [27]. To define sarcopenia we used the criteria from the European Working Group on Sarcopenia in Older People [28]. Functional performance was evaluated through Katz’s index for activities of daily living (ADLs) [29] and Lawton’s scale for instrumental activities of daily living (AIDL) [30].

Information on consumption of VD and other micronutrient supplements was obtained by a semi-quantitative FFQ during the last 7 days prior the survey. Those OA taking VD supplements were excluded from the analysis (*n* = 20).

### 2.3. Statistics

A total 783 OA were analyzed. Data are reported as frequencies for categorical variables and means, geometric means, and standard deviations for continuous variables. Bivariate analysis was conducted using a Fisher´s exact test for categorical variables and ANOVA test for continuous variables in order to compare the characteristics of OA by VA and VD status.

We adjusted for retinol, vitamin D, log-hepcidin, and ferritin levels by using inflammatory markers (AGP and CRP), given that inflammatory or infectious conditions could affect their serum levels. For calculating the adjusted- log hepcidin values (mean ± SE), VAD and VDD prevalence, we used a regression correction approach, applying marginal estimation [31].

To assess the independent association of 25(OH)D and retinol categories with hepcidin levels, we performed linear and ordinal regression analyses. Hepcidin was the outcome variable and was expressed using both the log of hepcidin and tertiles of hepcidin levels. Since hepcidin levels were non-normally distributed, their values were logarithmically transformed. In the linear models we used bootstrap-based clustered errors to account for the correlation between state-level observations [32]. 

Independent variables (VA and VD levels) were explored as follows: by unit of change, by a change in −10 units, using log transformed VA or VD levels, and by categorical status (VAD and VDD), where those with 25(OH)D levels ≥50nmol/L and retinol ≥20ug/dL were considered the reference. 

For each outcome variable (log and tertiles of hepcidin), we fitted a set of models: Model 1: adjusted for sex, age, indigenous, socioeconomic status, vitamin A or vitamin D status, and status of inflammation (AGP and CRP); Model 2: adjustments made in model 1 plus ferritin, sTFR, BMI, renal disease, anemia, frailty, current smoking, and AINES use; and Model 3: same as model 2, plus IL-6.

## 3. Results

Characteristics of 783 OA are presented in Table 1. Overall, 60.2% were women, 33% had CRP values higher than 5mg/L, 7.7% had AGP values higher than 1g/dL, 41.8% had a functional disability as defined by IADL, 35.9% had anemia, 9.4% had VDD, 2.4% had VAD after correcting for inflammation, 5.1% were ID, 9.2% were B12D.

Table 2 shows the characteristics of OA by VD and VA status. Those with VDD were more likely to be female, ages 80 and older, and suffer from anemia (multiple causes), sarcopenia, diabetes, functional disability, frail, use of medication, higher serum IL-6, creatinine, and eGFR values and lower hemoglobin values as compared to those OA with 25(OH)D levels ≥ 50 nmol (*p* < 0.05). Those with VAD were more likely to be of lower SES, have low prevalence of VDD, and suffer from anemia (due to inflammation and renal disease), low serum iron, high ferritin, higher status of inflammation, normal BMI, sarcopenia, cirrhosis, higher medication consumption, and higher serum levels of IL-6, CRP, EPO and lower Hb values in comparison with those OA with normal retinol levels (*p* < 0.05). 

Log-normalized hepcidin levels adjusted by age, sex, AGP, and CRP, were not statistically significantly different between those with normal levels of VD (25(OH)D ≥50 nmol) (2.4 ± 0.18) versus those with VDD (25(OH)D <50 nmol) (2.3 ± 0.26; *p* = 0.061) (Figure 1a). As hypothesized, serum hepcidin levels were significantly higher in OA with VAD than in OA with normal serum retinol (2.8 ± 0.18 vs. 2.4 ± 0.35, *p* = 0.043) (Figure 1b).

When looking at the descriptive association between biomarkers and VDD and VAD status, the log normalized levels of retinol were correlated with biomarkers of inflammation, such as s-IL6 (rho = −0.33, *p* < 0.001) and CRP (rho = −0.26, *p* < 0.001); but not with AGP (rho = 0.05, *p* = 0.14). For VD, 25(OH)D had a weak negative correlation with IL-6 (IL-6 rho = –0.08, *p* = 0.02), but was not significant for CRP (rho = 0.006, *p* = 0.84) or AGP (rho = –0.02, *p* = 0.53). When stratifying by anemia, the correlation between retinol and biomarkers was stronger for anemics (rho = −0.39, *p* < 0.001 for IL6; rho = −0.30, *p* < 0.001 for CRP; and rho = 0.06 *p* = 0.28 for AGP) than non anemics (rho = −0.28, *p*<0.001 for IL6; rho = −0.22, *p* < 0.001 for CRP; and rho = 0.05, *p* = 0.20 for AGP). For 25(OH)D levels, correlation with biomarkers in non-anemics: rho = −0.13, *p* = 0.003 for IL6; rho = –0.08, *p* = 0.07 for CRP; and rho = −0.03, *p* = 0.46 for AGP; in anemics: rho = 0.02, *p* = 0.66 for IL6; rho = 0.13, *p* = 0.02 for CRP; and rho = 0.006, *p* = 0.911 for AGP. 

Figure 2 demonstrates the positive correlation between log retinol and serum iron levels (rho = 0.20, *p* < 0.001) (Figure 2a); while logged 25(OH)D levels were not correlated with log serum iron levels (rho = 0.03, *p* = 0.40) (Figure 2b).

Table 3 shows that retinol values in all presentations (non-transformed, logarithmically transformed, and grouped by 10 units or VAD status), were inversely associated with log-hepcidin levels in the adjusted models. In the adjusted model 1, those with VAD has an increased odds of higher hepcidin levels (coefficient = 0.43, 95%CI: 0.15, 0.7) (Model 1, for VA deficiency and Log of hepcidin). In the ordinal model using tertiles of hepcidin levels as outcome variable, OA with VAD had a higher probability of falling within the highest tertile of hepcidin (OR = 2.15, 95%CI: 1.24, 3.74) (Model 2, for VA deficiency, ordinal regression model). Model 3 considered the adjustment by IL-6; although the association previously observed diminished in magnitude, VA still remained significantly and inversely associated with hepcidin levels (*p* < 0.05) (Table 3). 

Neither 25(OH)D nor VD status were associated with hepcidin levels. The adjusted models did not show a significant association of VD and hepcidin levels in this population (*p* > 0.05) (Table 3).

## 4. Discussion

In this study, we found that retinol levels were associated with hepcidin concentrations, after adjusting for the inflammatory condition, since VAD were associated with higher hepcidin levels in OA. This association occurred in a population living in the southern region of Mexico, where the prevalence of anemia is high, mainly caused by inflammation [19]. Meanwhile, 25(OH)D, was not significantly associated with hepcidin levels in this population.

Previous analysis showed that VAD and VDD were both associated with anemia but through different etiologies. VAD but not VDD showed a strong association to anemia of inflammation in this population, after considering confounders. VDD was associated with anemia of nutritional deficiencies and multiple causes. Although a misclassification can occur because the criteria used [19], all these results may suggest that vitamin A may play a role in the pathophysiology of AI, with hepcidin as an important mediator.

The regulation of hepcidin synthesis is stimulated by iron overload, hypoxia, erythropoiesis and inflammation. In the setting of inflammation, hepcidin expression is driven mainly by an increase in the pro-inflammatory interleukin-6, which occurs via the JAK-STAT3 pathway [33]. The resulting higher hepcidin levels cause iron retention in macrophages, hepactocytes, and enterocytes with reduced availability of iron for erythropoiesis, impairing the proliferation of erythroid progenitor cells affecting the heme synthesis [34].

In humans, the ability of both (VA and VD) liposoluble vitamins to modulate hepcidin expression is unknown; nevertheless, it is well documented that retinol and vitamin D interact due to their role as ligand-regulated transcription factors for multiple gene expression in a wide range of biological processes (i.e., systemic inflammation) [35,36]. Retinoid acid receptors (RARs) and retinoid X receptors (RXRs) are transcriptional factors of the nuclear receptor (NR) superfamily. RARs and RXRs form RAR/RXR heterodimers and bind to RARE’s located in the gene promoters, modulating expressions of target genes through activation by their cognate ligands [35]. RXRs may also heterodimerize with other members of the NR, as the nuclear vitamin D receptor (VDR), binding to VD3 response elements (VDREs) in the promoters of VD3-responsive genes [36]. According to Bacheta et al., hepcidin have VDREs in the promoter region of *HAMP* gene in monocytes [8]. This may suggest that the interaction between these two vitamins may activate or repress hepcidin expression. In our study, OA did not share both vitamin deficiencies (with the exception of two OA). Therefore, it was not possible to explore the synergistic interaction between both vitamin deficiencies with hepcidin levels. 

Most of the experimental evidence suggests that both VA and VD may downregulate hepcidin expression in an independent way of their suppression activity of inflammatory pathways. The evidence regarding the effect of VAD on hepcidin expression came from experimental studies in rodents. Some authors have found a direct link between VA depletion and higher *HAMP* expression. Arruda et al. [14] first showed that VAD rats had increased liver hepcidin mRNA level, higher iron spleen concentration and a higher oxidative status than control rats. In 2012, Citelli et al. [13] found that VAD mice, liver hepcidin, and ferritin mRNA levels were upregulated, suggesting that VAD affected hepatic iron mobilization as well the transcription factors of protein genes involved in the iron bioavailability, but without affecting iron absorption process. In contrast, Da Cunha et al. [37] found that VAD rats had a lower hepatic HAMP mRNA levels to half those of the control levels and suggested that VAD modulates iron metabolism via ineffective erythropoiesis by down-regulated renal Epo mRNA and up-regulated Hmox1 in the spleen [37]. By using the same protocol in rats, Ribeiro Mendez et al. [38] found that VAD up-regulated hepatic Bmp6 and Hfe mRNA levels and down-regulated hepatic HAMP mRNA levels, impairing HJV-BMP6-SMAD signaling pathway that normally activates the expression of hepcidin in ID. Differences in the results between these two studies may be due in part to the rats’ diet treatment, as well as the baseline levels of VA stored in their livers.

In populations with a high risk of anemia and micronutrient deficiency, vitamin A supplementation (VA-Sup) has been suggested as a strategy for improving Hb levels and ameliorating anemia. A recent meta-analysis of VA-Sup on anemia and iron status in different life stages (not including OA) showed that VA-Sup reduced the risk of anemia by 26% by improving hemoglobin and ferritin levels in individuals with low serum retinol (but without effects on the prevalence of ID by using serum ferritin as biomarker) [15]. In children with ID and VAD, VA-Sup increased erythropoiesis by mobilizing iron stores without changing the ratio of transferrin receptor to serum ferritin [16]. Even though the exact mechanism of VA is unknown, diminishing Hb through an elevation in hepcidin levels appears to be one of several biological pathways by which VA status can induce anemia. The profile of OA with VAD in our study is characteristic of functional iron deficiency due to inflammation that lead to AI. The pro-inflammatory cytokines affect the erythropoiesis causing EPO resistance [39]. It is possible that higher EPO levels in the VAD group be a consequence of EPO resistance due to chronic inflammation [40]. Due to their characteristics, this group of OA (with VAD) seems to have a higher risk of health adverse events [41,42].

Contrary to our hypothesis, VDD was not associated with hepcidin levels in our sample, even after stratifying by anemia (data not shown). Few experimental studies in cell lines and pilot studies of VD supplementation (VD-Sup) in humans have shown a link between hepcidin and VD. Bacchetta et al. [8] identified a VDRE binding site on human hepcidin promoter and 1,25-dihydroxyvitamin D (1,25(OH)2D3) directly downregulated *HAMP* gene transcription (by 0.5-fold) and ferritin, while increasing expression of ferroportin. A pilot study of a single oral dose of VD2 (100,000 IU) in seven healthy adults showed that VD-Sup increased serum 25D-hydroxyvitamin D (25(OH)D)] and decreased hepcidin levels by 34% within 24 h. Zughaier et al. [11] showed that (1,25(OH)2D3) was associated with reduced hepcidin expression and increased ferroportin and NRAMP1 expression in vitro and in vivo in inflammatory conditions by regulation of hepcidin-ferroportin axis in macrophages. A randomized, double-blind, placebo-controlled trial pilot study of VD-sup (*n* = 38, cholecalciferol, 50,000 IU weekly for 12 weeks) showed an increase in serum 25(OH)D and a decrease in serum hepcidin in subjects with early stage CKD [11]. Smith et al.’s study [12] demonstrated a 73% decrease of plasma hepcidin in 28 healthy adults treated with vitamin D3 compared with placebo, after a single oral dose of 250,000 IU for one week. In contrast, Panwar et al. [43] did not find any association of VD-Sup on hepcidin levels in 40 adults with CKD (stage 3 or 4) by using a randomized, placebo controlled double-blinded study design (calcitriol 0.5 mcg daily for six weeks). A distinction from the other studies is that CKD patients were eligible irrespective of their baseline vitamin D status, and the authors discuss that the effects of VD-Supl on hepcidin are more robust in VDD status. Observational studies that explore the association of VDD on hepcidin levels are scarce across all population groups. Recently, lower hepcidin levels were associated with higher 25(OH)D status in children with inflammatory bowel disease [44]. In our study, OA were ambulatory, and they had diverse chronic comorbidities with a diverse pro-inflammatory profile, different from those with autoimmune disease. 

The results from our study are likely different from other VD studies due to a number of factors. First, the target population. We are analyzing OA with a high prevalence of anemia and inflammation (~40%), in comparison to healthy, young adults with a high prevalence of VDD that respond to VD-Supl or with early stages of CKD. Second, the inflammatory response is different in OA versus healthy young people due to dysregulation of the immune system, the immunesenescence and inflammation that occurred in the course of normal aging (increased levels 2–4-fold of pro-inflammatory cytokine and decreased level of anti-inflammatory cytokine in comparison with young adults) [3,4,45], as well as the frailty condition [46] and metaflammation [5,47] (due to high prevalence of chronic comorbidities like DM, dyslipidemia, and hypertension). In addition, most of these studies highlight the VD-Sup effect over hepcidin levels, rather than basal 25(OH)D levels. Third, VDD was not as prevalent in this population as OA from other national surveys [10,17,48], likely because sun exposure in the southern region latitude is higher in comparison with other regions of Mexico. A larger sample or higher prevalence of VDD might have altered the results.

Fourth, we used 25(OH)D levels to determine VD status, and it is possible that the active form of VD [1,25(OH2)D3], could be a better indicator of their activity as a suppressor of immune systemic inflammation [9,49], despite the fact that 25(OH)D is the accepted standard biomarker for assessing VD status. 

Our results were based-population of OA from urban areas in a region where predominate multiple nutritional deficiencies, anemia, chronic diseases and obesity in all age groups [50]. Regional differences may account for differences in ethnicity, environment, culture, and sociodemographic characteristics in comparison with other regions of Mexico and from other OA populations. 

No studies have previously documented the association between VAD and hepcidin in OA, largely because VAD only continues to be a public health issue in developing countries. Even when prevalence of VAD was lower than VDD, the association was significant mainly because serum retinol is the active form of VA and homeostatically controlled that drop until liver reserves are very low [21], while for VD status, the active form 1, 25(OH2)D3 was not measured. Higher levels of hepcidin concentration in OA may be the result of multiple stimulus rather than one cause (iron overload plus inflammation, immunosenescence, metaflammation) and the lack of association with 25(OH)D status in optimum levels could be the result of a reduced renal function or a failure of incorporation into the cell [2]. Some polymorphism associated with biomarkers of inflammation, as well as those for the VD pathway, may better reflect the risk rather than serum baseline levels, since the prevalence of these polymorphs may vary across different ethnicities and populations [51]. The polymorphism associated with VDR in the Mexican population is higher in the southern region of Mexico (30%), being present in the 29% of Mayan ethnic population [52]. This risk profile may explain, that even in normal 1,25(OH2)D3 and 25(OH)D serum concentration, VD is not properly used by the cell to execute their multiple action.

One limitation of the present study is the cross-sectional design that limits the ability to infer causation. The significant association between retinol and hepcidin levels in the adjusted models (for inflammation), may be the result of residual confounding due to serum retinol drops under inflammatory conditions. Retinol levels might indicate the nutritional status of VA, or be the result of an inflammatory effect. Adjusting for biomarkers of inflammation is a helpful strategy to address confusion. Nevertheless, when we explored the correlation between intake of retinol equivalents from the diet and retinol serum in the OA from this study, the correlation was positive (Rho = 0.11, *p* < 0.001); indicating that retinol values and VAD status is not entirely explained by inflammation, may also reflect the nutritional status of VA.

Additionally, we included adjusted for IL-6 in the model, a known mediator in the association, despite the fact we did not use structural models. As result, the magnitude of association was reduced but still remained statistically significant. In addition, reverse causality is common in biomarker measurement and, therefore, VAD and VDD may be the result of chronic inflammatory pathways that underlie the higher hepcidin levels. Moreover, the sample was restricted to OA from urban areas and they were not representative of OA from the southern region nor from the State of Campeche nor Yucatan. Our sample of OA with VAD had limited power and thus we were not able to explore stratified analysis for anemia condition and other covariables.

In summary, the results of the present study shown that VA, yet not VD, is inversely associated with hepcidin levels in OA with a high prevalence of anemia mainly due to inflammatory etiology. Further longitudinal studies must explore the temporal trends between baseline levels of hepcidin, VD, and VA status in settings where AI in OA is highly prevalent.

## 5. Conclusions

This is the first study that analyzed the association between serum retinol status, serum 25(OH)D and hepcidin levels in older adults. This data indicated that serum retinol levels, but not 25(OH)D, was inversely associated to serum hepcidin concentration, after considering the inflammatory and anemia status. These findings have a public health implication since anemics had higher prevalence of VAD and VDD than non-anemics.

## Figures and Tables

**Figure 1 nutrients-11-00988-f001:**
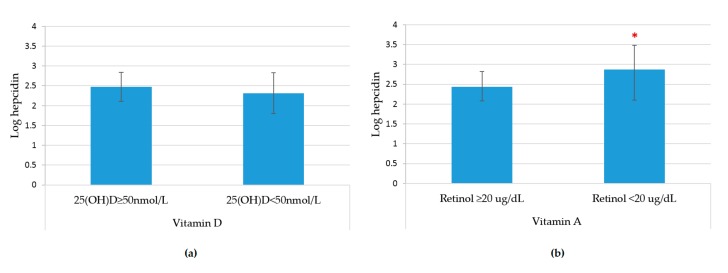
Logarithmic normalized hepcidin levels by vitamin D and vitamin A status, adjusted by age, sex and status of inflammation (CRP and AGP) in Mexican older adults. (**a**) Mean of log-hepcidin by vitamin D deficiency; and (**b**) mean of log hepcidin by vitamin A deficiency. * *p* < 0.05.

**Figure 2 nutrients-11-00988-f002:**
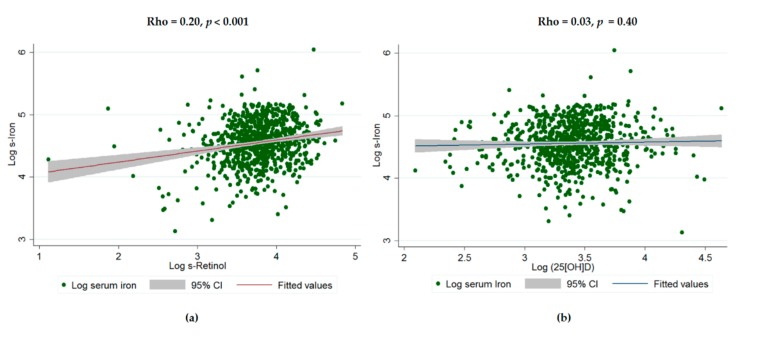
Smoothed correlation of serum retinol and 25(OH)D with serum iron in OA. (**a**) Correlation between log-serum retinol and log-serum iron; and (**b**) correlation between log-serum 25(OH)D and log-serum iron.

**Table 1 nutrients-11-00988-t001:** Descriptive characteristics of older Mexican adults sample.

Characteristic	Sample (*n*)	Frequency (%)
**Sex (women)**	471	60.2
**Age group (years)**		
60–69	388	49.6
70–79	269	34.4
80+	126	16.1
**Speak an indigenous language (yes)**	264	33.7
**Tertile of SES (asset index)**	777	
Tertile 1	261	33.6
Tertile 2	269	34.6
Tertile 3	247	31.8
**Body Mass Index**	770	
Normal	176	22.9
Overweight	291	37.8
Obesity	303	39.4
**CRP (>5 mg/L)**	264	33.7
**AGP (>1 g/dL)**	60	7.7
**IL6 (>10 pg/mL)**	85	10.9
**Status of inflammation**	783	
Reference	501	63.9
Incubation	222	28.3
Early convalescence	42	5.3
Late convalescence	18	2.3
**Functional disability**	783	
ADL	231	29.5
IADL	327	41.8
**Frailty**	783	
Not frail	314	40.1
Pre-frail	365	46.6
Frail	104	13.3
**Sarcopenia**	124	16.3
**Medical condition**	783	
T2 Diabetes	238	30.4
Hypertension	391	49.9
Dyslipidemia	273	34.8
Arthritis	95	12.1
Cirrhosis	157	20.1
Cancer	31	4
Chronic kidney disease ^1^	187	23.8
**Use of medication**	574	73.3
**NSAID consumption**	120	15.3
**SAID consumption**	20	2.6
**Current Smoking**	195	24.9
**Anemia**	281	35.9
**Anemia causes**	783	
No anemia	502	64.11
Renal	81	10.34
Nutritional	22	2.81
Inflammation	41	5.24
Multiple causes	9	1.15
Unexplained	128	16.3
**Serum micronutrient deficiency ^2^**		
B12 deficiency	72	9.2
Iron deficiency	40	5.1
Vitamin A Deficiency	27	3.4
Adjusted Vitamin A Deficiency	27	2.4
Vitamin D Deficiency	74	9.5
Adjusted Vitamin D Deficiency	74	9.4
**Means of serum biochemical parameters** **^3^**		
Hemoglobin (g/dL)	783	12.8 ± 1.7
CRP (mg/L)	783	3.3 ± 3.3
AGP (g/dL)	783	0.5 ± 1.5
IL-6 (pg/mL)	783	2.7 ± 3.7
EPO (mUI/mL)	783	11 ± 1.6
Creatinine (mg/dL)	783	0.9 ± 0.6
Estimated filtration glomerular rate by CKD-EPI Creatinine (mL/min/1.73 m^2^)	783	78.9 ± 17.6
Hepcidin (ng/mL)	783	12.2 ± 3

Abbreviations: ADL: Basic activity of daily living; IALD: Instrumental Activity of Daily Living; NSAID: Non-steroid anti-inflammatory drugs; CRP: C reactive protein; AGP: Alpha glycoprotein 1 acid; IL-6: interleukin-6; EPO: Erythropoietin; CKD-EPI: Estimated glomerular filtration rate (eGFR) by the Chronic Kidney Disease Epidemiology Collaboration (CKD-EPI) equation. ^1^ Chronic kidney disease was defined if eGFR by CKD-EPI creatinine was <60 mL/min/1.73 m^2^ [25] or a previously-diagnosed kidney disease by a physician. ^2^ Vitamin B12 deficiency was defined as <−0.5 SD, considering homocysteine and folate concentration according to Fedosov’s equation [24]. Iron deficiency was defined as serum ferritin <15 ng/mL after correcting for inflammation according Turnham [23] or if sTfR >28 nmol/L. Vitamin A deficiency (VAD) if serum retinol <20 μg/dL [21]. Vitamin D deficiency (VDD) if 25(OH)D <50 nmol/L [20]. VAD and VDD prevalence were adjusted considering stage of inflammation (AGP and CRP levels) by regression approach. ^3^ Mean ± SD for Hb, creatinine and eGFR and geometric mean for CRP, AGP, and IL-6.

**Table 2 nutrients-11-00988-t002:** Frequency of characteristics of Mexican older adults by vitamin D and vitamin A status.

	Vitamin D			Vitamin A		
	25(OH)D ≥50 nmol/L	25(OH)D <50 nmol/L	*p* Value *	Retinol ≥20 μg/dL	Retinol <20 μg/dL	*p* Value *
	*n* = 709 %	*n* = 74 %		*n* = 756 %	*n* = 27 %	
**Sex (women)**	58.1	79.7	<0.001	60.6	48.1	0.231
**Indigenous**	33.4	36.5	0.607	32.5	66.7	0.001
**Age group (years)**						
60–69	51.2	33.8		50	37	
70–79	33.4	42.2		34.3	37	
80 and older	15.4	23	0.012	15.7	25.9	0.253
**Tertile of SES (asset index)**						
Tertile 1	33.4	35.1		32.5	63	
Tertil 2	35.0	31.1		35.1	22.2	
Tertil 3	31.6	33.8	0.792	32.4	14.8	0.007
**Anemia causes**						
No anemia	65.4	51.4		65.2	33.3	
Renal	9.9	14.9		10.1	18.5	
Nutritional	5.2	5.4		2.8	3.7	
Inflammation	5.2	5.4		4.4	29.6	
Multiple causes	0.7	5.4		1.2	0	
Unexplained	16.2	17.6	0.008	16.4	14.8	<0.001
**Vitamin D status**	-	-	-			
25(OH)D ≥75 nmol/L	-	-	-	48.4	74.1	
25(OH)D 50–74 nmol/L	-	-	-	42.1	18.5	
25(OH)D <50 nmol/L	-	-	-	9.5	7.4	0.025
**Retinol <20 ug/dL**	3.5	2.7	0.99	-	-	-
**Vitamin B12 deficiency**	10.1	16.2	0.119	10.4	18.5	0.202
**Iron deficiency**	4.8	8.1	0.218	5	7.4	0.643
**Low s-ferritin (<15 ng/mL)**	1.4	2.7	0.316	1.6	0	0.99
**Low serum iron (<60 ug/dL)**	8.5	9.5	0.826	7.5	37	<0.001
**High s-ferritin (≥350 ng/mL)**	7.5	5.4	0.643	6.7	22.2	0.010
**CRP (>5 mg/L)**	34	31.1	0.699	32.5	66.7	0.001
**AGP (>1 g/L)**	7.5	9.5	0.494	7	25.9	0.003
**IL6 (>10 pg/mL)**	10.3	16.2	0.119	9.3	55.6	<0.001
**Status of inflammation**						
Reference	63.6	67.6		65.1	33.3	
Incubation	28.9	23		27.9	40.7	
Early convalescence	5.1	8.1		4.6	25.9	
Late convalescence	2.4	1.4	0.488	2.4	0	<0.001
**Body Mass Index**						
Normal	22.8	23.2		21.9	50	
Overweight/Obese	77.2	76.8	0.99	78.1	50	0.003
**Sarcopenia**	15	29.4	0.005	15.5	38.5	0.005
**Comorbidities previously diagnosed by a physician**						
Type 2 Diabetes	29.2	41.9	0.033	30.4	29.6	0.99
Hypertension	48.9	59.5	0.089	50.1	44.4	0.696
Dyslipidemia	34.3	40.5	0.306	35.6	14.8	0.025
Cancer	3.8	5.4	0.524	4.1	0	0.620
Cirrhosis	2.1	4.1	0.238	1.9	14.8	0.002
Chronic kidney disease ^1^	22.9	32.4	0.085	23.7	29.6	0.492
Arthritis	20	20.3	0.99	20.5	7.4	0.139
**Functional Disability**						
ALD	27.4	50	<0.001	29.2	37	0.394
IALD	39.6	62.2	<0.001	41.3	55.6	0.165
**Frailty condition**						
No Frail	42.2	20.3		40.5	29.6	
Pre-frail	47	43.2		46.8	40.7	
Frail	10.9	36.5	<0.001	12.7	29.6	0.058
**Current smoking**	24.9	24.3	0.99	24.8	25.9	0.825
**Use of medication**	72.2	83.8	0.038	74.1	51.9	0.015
**NSAID**	14.5	22.9	0.062	15.6	7.4	0.411
**SAID**	2.4	4	0.425	2.5	3.7	0.509
**Means of serum biochemical parameters** ^2^						
Hemoglobin (g/dL)	12.8 ± 1.7	12.1 ± 1.7	0.001	12.8 ± 1.7	12 ± 1.7	0.016
CRP (mg/L)	3.3 ± 3.3	3.3 ± 3	0.855	3.3 ± 3	10 ± 5.5	<0.001
AGP (g/dL)	0.5 ± 1.5	0.6 ± 1.5	0.269	0.5 ± 1.5	0.5 ± 2.2	0.41
IL-6 (pg/mL)	2.5 ± 3.7	3.7 ± 3	0.016	2.5 ± 3.3	13.5 ± 3.3	<0.001
EPO (mUI/mL)	11 ± 1.6	11 ± 1.6	0.269	10 ± 1.6	14.9 ± 2	<0.001
Creatinine (mg/dL)	0.9 ± 0.5	1.1 ± 1.4	0.003	0.9 ± 0.7	0.9 ± 0.3	0.795
Estimated filtration glomerular rate by CKD-EPI Creatinine (mL/min/1.73 m^2^)	79.5 ± 16.9	72.7 ± 23.1	0.002	78.9 ± 17.5	77.4 ± 21.8	0.668

Abbreviations: CRP: C reactive protein; AGP: Alpha glycoprotein 1 acid; IL-6 Interleukin 6; ADL: Basic Activity of daily living; IALD: Instrumental Activity of Daily Living; NSAID: Non-steroid anti-inflammatory drugs; SAID: Steroid anti-inflammatory drugs; EPO: Erythropoietin; CKD-EPI: Estimated glomerular filtration rate (eGFR) by the Chronic Kidney Disease Epidemiology Collaboration (CKD-EPI) Equation. ^1^ Chronic kidney disease was defined if estimated glomerular filtration rate by CKD-EPI creatinine was <60 mL/min/1.73 m^2^ [25] or a previously-diagnosed kidney disease by a physician. ^2^ Mean ± SD for Hb, creatinine and eGFR and geometric mean for CRP, AGP and IL-6. * Fisher’s exact test for categorical variables and ANOVA test for continuous variables.

**Table 3 nutrients-11-00988-t003:** Adjusted linear and ordinal regression model for Vitamin A and D status and its association with log-hepcidin levels in OA.

	Model 1, *n* = 777		Model 2, *n* = 765		Model 3, *n* = 765	
**Vitamin A**						
**Outcome: Log of hepcidin**	**β**	**95CI%**	**β**	**95CI%**	**β**	**95CI%**
Retinol (μg/dL)	−0.003	(−0.004, −0.003)	−0.004	(−0.004, −0.003)	−0.003	(−0.004, −0.002)
Log retinol	−0.14	(−0.19, −0.1)	−0.15	(−0.2, −0.09)	−0.1	(−0.13, −0.06)
Decrement (10 u)	0.03	(0.03, 0.04)	0.04	(0.03, 0.04)	0.03	(0.02, 0.04)
VA deficiency	0.43	(0.15, 0.7)	0.35	(0.09, 0.62)	0.24	(0.08, 0.39)
**Outcome: Tertile of hepcidin**	**OR**	**95CI%**	**OR**	**95CI%**	**OR**	**95CI%**
Retinol (μg/dL)	0.994	(0.993, 0.996)	0.99	(0.99, 0.995)	0.99	(0.993, 0.997)
Log retinol	0.77	(0.76, 0.78)	0.77	(0.73, 0.81)	0.83	(0.82, 0.84)
Decrement (10 u)	1.06	(1.05, 1.07)	1.07	(1.05, 1.08)	1.05	(1.03, 1.08)
VA deficiency	2.29	(1.36, 3.88)	2.15	(1.24, 3.74)	1.82	(1.23, 2.69)
**Vitamin D**						
**Outcome: Log of hepcidin**	**β**	**95CI%**	**β**	**95CI%**	**β**	**95CI%**
25(OH)D (ng/mL)	0.004	(−0.002, 0.01)	0.004	(−0.001, 0.009)	0.004	(−0.002, 0.01)
Log vitamin D	0.17	(−0.04, 0.38)	0.16	(−0.02, 0.34)	0.16	(−0.03, 0.35)
Decrement (10 u)	−0.04	(−0.11, 0.02)	−0.04	(−0.09, 0.01)	−0.04	(−0.1, 0.02)
VD deficiency	−0.14	(−0.33, 0.04)	−0.18	(−0.43, 0.06)	−0.19	(−0.4, 0.02)
**Outcome: Tertile of hepcidin**	**OR**	**95CI%**	**OR**	**95CI%**	**OR**	**95CI%**
25(OH)D (ng/mL)	1.01	(0.99, 1.02)	1.003	(0.99, 1.01)	1.003	(0.99, 1.02)
Log vitamin D	1.25	(0.78, 1.99)	1.17	(0.78, 1.75)	1.17	(0.75, 1.83)
Decrement (10 u)	0.95	(0.83, 1.09)	0.97	(0.87, 1.09)	0.97	(0.85, 1.11)
VD deficiency	0.79	(0.53, 1.17)	0.74	(0.42, 1.28)	0.73	(0.44, 1.23)

Model 1 adjusted for sex, age, indigenous, tertile of socioeconomic status, vitamin A or vitamin D status and status of inflammation (CRP and AGP). Model 2 adjusted for model 1 plus ferritin, transferrin receptor, current smoking, body mass index, chronic renal disease, anemia, frailty, AINES and AIE consumption. Model 3 adjusted for model 2 plus IL-6.

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
