# Peer review of "Serum Retinol but Not 25(OH)D Status Is Associated With Serum Hepcidin Levels in Older Mexican Adults"

_nutrients, 2019, doi:10.3390/nu11050988_

Round 1
Reviewer 1 Report
De la Cruz-Góngora et al. examined correlation between vitamin A (VA), vitamin D (VD) and hepcidin in older Mexican adults (OA). VA efficiency (VAD) and VD deficiency (VDD) were present in 3.2 % and 9% in OA. VAD but not VDD showed a higher probability for higher hepcidin. They concluded that VAD but not VDD status was inversely correlated with hepcidin in OA, independent of inflammation.
General comment:
The study is interesting but there are some concerns. First, OA with various clinical conditions that differentially affect hepcidin are included. If inflammation is defined by CRP>5mg/L and AGP>1g/L. [ref23, 30] as the authors suggested, 41.3% of OA (Table 1) have inflammation that can induce VAD, VDD, high hepcidin and ferritin, leading to functional iron deficiency (FIDA). Cirrhosis induces VAD, VDD and decreases hepcidin. Dislipidemia differentially affects hepcidin but what type of dyslipidemia is not given. CKD with low GFR is associated with VDD and high inflammation, and low GFR by itself increases hepcidin due to its low urinary excretion. Furthermore, some (2.8%) OA received VD supplements.
Second, ferritin and retinol are adjusted for inflammation according to ref30 but this method is based on preschool children and reproductive women but not OA. Is this method applicable to OA? In addition, VD and hepcidin are not adjusted for inflammation. Third, like inflammation, malnutrition decreases VA and VD and increases hepcidin and ferritin, leading to FIDA. In this study, VB12 is only measured as a nutritional marker but serum albumin, leptin and total cholesterol are not evaluated. The effect of VB12 deficiency on hepcidin remains unknown although it did not affect hepcidin in children [Adv Clin Exp Med. 2017, 6, 621-5]. Fourth, OA with VAD have higher inflammation, obesity and dyslipidemia that increase hepcidin as compared to those without. However, there was no difference in these factors between OA with and without VDD. Since VD supplements reduce inflammatory markers and hepcidin, VD supplements may affect the results and thus patients receiving VD supplements should be excluded for analysis. Finally, the number of OA with minor inflammation was not given. If VAD increases hepcidin independent of inflammation, the study should be performed using OA without inflammation to examine whether VAD induces hepcidin and affects iron metabolism independent of inflammation process. Taken together, this study does not exclude the possibility that higher hepcidin may be due to high inflammation but not due to VAD in OA.
Minor comment:
Introduction
1. Line 59-61, line 70 and line 215. This reference [19] is not published and should be deleted.
Methods
1. Line 67. The authors recruited 829 OA (ages ≥ 60y) to understand the causes of anemia. However, only 35.5% of participants have anemia (Table1). Criteria for selection of OA are unclear and should be clarified.
2. Line 77. There is a circadian rhythm in VD levels [Eur J Endocrinol. 2002,146, 635-42]. This may affect the results. Is the time of blood sampling similar in all OA?
3. Line 80, 91-92. Data for homocysteine and EPO are not given. In addition, data for creatinine and GFR should be given, regardless of CKD. Is there any CKD patient on dialysis?
4. Line 101. “mcg/dL” should be “mg/dL”.
Results
1. Data for Hb should be given in Tables 1 and 2. Is there any correlation between Hb and ferritin, hepcidin, VA, VD, CRP, AGT, IL-6, creatinine and GFR?
2. While percentage of anemia is very high (66.7%) in OA with VAD than in those without (34.7%, Table2), percentage of anemia in a whole group is 35.5% (Table 1). It is hard to understand the data only by frequency. The actual number of the patients in each category should be given in Tables 1 and 2. In Table 1, the number of the patients is 829 but it should be 803 as described in line 71. In Table 2, at least the number of the patients in each group, namely VD≥50nml/L, VD<50nmol/L, retinol≥20mg/dL and retinol<20mg/dL should be given.
3. Table 2 only shows percentage of anemia but what type of anemia is not clarified.
4. Line 159-162. Prevalence of hypertension in OA with and without VDD is not significant (P>0.05, Table 2).
5. Line 162-165. OA with VAD had low prevalence of VDD. Actual number of the patients with VDD (VD<50nmol/L) should be given in Table 2.
6. Line 159-165. The authors stated that anemia is more prevalent in OA with VDD and VAD than those without. Table 2 shows high prevalence of VB12 deficiency that causes anemia in these groups. Does it mean that malnutrition is more prevalent in OA with VAD and VDD than those without? Or. Does it mean that prevalence of VB12 deficiency anemia is higher in OA with VAD and VDD than those without?
7. Results show no difference in iron deficiency and low ferritin in OA regardless of VD and VA status, whereas anemia of inflammation is only higher is OA with VAD but not OA with VDD. The authors stated that adjusting for VD supplements did not change the results [line299] but they did not examine the data when such patients receiving VD supplements are excluded. In addition, actual number of OA with VDD (78 patients) is higher that of those with VAD (27 patients) and thus statistical significance is stronger in the former group than in the latter group. It is not clear how lower prevalence of OA with VDD in this study than in other countries [line300-] can explain the reason why VDD is not associated with higher hepcidin in this study.
8. Prevalence of anemia of inflammation is only higher in OA with VAD than those without. However, its prevalence is not different between OA with and without VDD. Prevalence of low serum iron, high ferritin, parameters of inflammation (CRP, IL-6 and AGP), obesity and comorbidities associated with inflammation are not different between OA with and without VDD. Only difference is found in prevalence of VB12 deficiency in OA with VDD than those without. Prevalence of anemia is higher in OA with VDD than those without. Does it mean that VB12 deficiency anemia is more prevalent in OA with VDD than those without?
9. Line 170-174. The authors stated that serum hepcidin levels were significantly lower in OA with VAD than in OA with normal retinol (Fig1a). However, this should be “serum hepcidin levels were significantly higher in OA with VAD than in OA with normal serum retinol (Fig1b).” Namely, Fig1a should be Fig1b and Fig1b be Fig1a.
10. The definition of inflammation (CRP>5mg/L, AGP1g/L) according to ref 23 and 30 may be used in this study. However, CRP was divided into 3 groups in Table 2; 0-3, 3-10 and >10mg/L instead of 0--5 and >5mg/L as shown in Table1. Why IL-6 levels were divided into ≤10 and >10 pg/mL? What are the normal values of IL-6? The definition of inflammation as measured by CRP, IL-6 and AGP should be clearly described.
11. Percentage of OA receiving drugs is different from that of those with NSAID plus SAID. Why do OA with VAD have lower prevalence of drugs (other than NSAID or SAID) as compared to those without?
12. High levels of ferritin and hepcidin and low levels of serum iron are characteristic of FIDA or anemia of inflammation. In this study, anemia, high inflammation, low serum iron and high ferritin and hepcidin, characteristic of FIDA, are associated with OA with VAD as compared to those without. Since hepcidin data are not adjusted for inflammation, this finding suggests that inflammation may induce VAD, leading to higher ferritin and hepcidin and subsequent anemia of inflammation. In fact, the levels of VA are inversely correlated with the levels of inflammatory markers (CRP, IL-6) and this negative correlation was stronger for anemics [line 179-189].
Discussion
1. Line 210-211. If so, same conclusion should be obtained in OA without inflammation.
2. Throughout discussion the authors refer to anemia but what type of anemia should be clarified.
3. Line 215-220. As described, ref19 should be deleted from the manuscript.
4. Line287. In this study, the authors stated that the effect of inflammation was adjusted for evaluation of hepcidin. However, they discussed that OA were associated with inflammatory condition possibly different from other study [ref40].
Figures
1. The name of variable should be indicated on vertical line.
Tables
1. Table 1 shows 829 OA but serum and hematological data are available in 803 OA (line 71).
2. Tables 1 and 2 should show raw data for hepcidin. CRP, IL-6, AGT, ferritin, EPO, serum albumin, leptin, total cholesterol, creatinine and GFR as well as the number of patients with CRP≥5mg/L and <5mg and those with and without high IL-6.
4. Why are p values missing in some variables of Table 2? The unit of VD in first raw of Table 2 should be “nmol/L”.
Miscellaneous
The authors should be careful about the use of abbreviations and the order of their appearance in the text.

Author Response
De la Cruz-Góngora et al. examined correlation between vitamin A (VA), vitamin D (VD) and hepcidin in older Mexican adults (OA). VA efficiency (VAD) and VD deficiency (VDD) were present in 3.2 % and 9% in OA. VAD but not VDD showed a higher probability for higher hepcidin. They concluded that VAD but not VDD status was inversely correlated with hepcidin in OA, independent of inflammation.
General comment:
The study is interesting but there are some concerns. First, OA with various clinical conditions that differentially affect hepcidin are included. If inflammation is defined by CRP>5mg/L and AGP>1g/L. [ref23, 30] as the authors suggested, 41.3% of OA (Table 1) have inflammation that can induce VAD, VDD, high hepcidin and ferritin, leading to functional iron deficiency (FIDA). Cirrhosis induces VAD, VDD and decreases hepcidin. Dislipidemia differentially affects hepcidin but what type of dyslipidemia is not given. CKD with low GFR is associated with VDD and high inflammation, and low GFR by itself increases hepcidin due to its low urinary excretion. Furthermore, some (2.8%) OA received VD supplements.
Response We agree with the comment. We have included OA with different health conditions that might affect hepcidin and VA/VD status due to inflammatory process; however, this is the actual situation of non-institutionalized OA in Mexico from the southern region (a less developed region), and it would be very difficult to find enough OA without any chronic condition nor inflammation to address the current research question. Furthermore, we considered all these variables as confounders in the regression models, excluding those OA who were using VD supplements.
Second, ferritin and retinol are adjusted for inflammation according to ref30 but this method is based on preschool children and reproductive women but not OA. Is this method applicable to OA?
Response We absolutely agree with this comment, in effect, the equation used by Turnham et al; does not considered OA population for adjusting ferritin values due to inflammation. Using the Turnham’s equation in this population, can result in residual confounding due to inflammation and ferritin values may not properly corrected at all for such effect, underestimating the ID status in this population. Because lack of studies (meta-analysis) that consider this adjustment in OA population, we decided to correct our values using Turnham’s equation rather than other methods (i.e. regression analysis, higher cut-off of ferritin to define ID).
In addition, VD and hepcidin are not adjusted for inflammation.
Response Considering that inflammation may confound the association explored between hepcidin and VA or VD status, we run again all the statistical models adjusting all models by AGP and CRP.
Usually, VD values and VDD prevalences have been reported in the literature without adjustment by inflammation. In our study, VD was not correlated to AGP nor CRP, then, there were no differences between the adjusted VDD prevalence and the non-adjusted VDD; nevertheless, this adjusted prevalence of VD for AGP and CRP was added in Table 1.
Hepcidin values reported in the figure and Table 2 were replaced by those corresponding to adjusted by CRP, AGP (figures and models). Unadjusted hepcidin values were only described in Table 1.
Third, like inflammation, malnutrition decreases VA and VD and increases hepcidin and ferritin, leading to FIDA. In this study, VB12 is only measured as a nutritional marker but serum albumin, leptin and total cholesterol are not evaluated.
Response Regarding the biomarkers mentioned above, leptin, total cholesterol and albumin were biomarkers not measured because those biomarkers were not considered in the original budget of the project. The frequency of dyslipidemia described, was self-reported if a physician previously had diagnosed it. (line 118).
The effect of VB12 deficiency on hepcidin remains unknown although it did not affect hepcidin in children [Adv Clin Exp Med. 2017, 6, 621-5].
Response Because there were no scientific evidence that B12 deficiency could affect hepcidin values in OA, this biomarker was not consider a confounder of the VA an VD association with hepcidin. Bivariate exploratory analysis, showed no correlation between B12 and hepcidin (p=0.410). For this reason, VB12 was not considered in the statistical models as covariable between the main associations explored.
Fourth, OA with VAD have higher inflammation, obesity and dyslipidemia that increase hepcidin as compared to those without. However, there was no difference in these factors between OA with and without VDD. Since VD supplements reduce inflammatory markers and hepcidin, VD supplements may affect the results and thus patients receiving VD supplements should be excluded for analysis.
Response We agree with this comment. VD supplementation may act as an effect modifier since it can modulate the immune and inflammatory responses and affect the results observed. For that reason, we exclude all OA that reported the consumption of VD supplements (n=20) and we run again all the analysis excluding them, with a final sample of 783 OA (Line 131-134). The main association did not changed nor the conclusion originally observed between VD status and hepcidin (table 3).
Finally, the number of OA with minor inflammation was not given.
Response We added (table 1 and 2), information on frequency of OA according to their inflammatory status (reference, incubation, early convalescence and late convalescence).
If VAD increases hepcidin independent of inflammation, the study should be performed using OA without inflammation to examine whether VAD induces hepcidin and affects iron metabolism independent of inflammation process. Taken together, this study does not exclude the possibility that higher hepcidin may be due to high inflammation but not due to VAD in OA.
Response We agree with the comment of the reviewer. In effect, the analysis showing the VAD and hepcidin association should considered those without inflammation. However, if we restrict the analysis to those without inflammation or minimal inflammation we found only 7 OA with VAD, the small number of OA with VAD and without inflammation constrains the possibility of conducting such analysis. We recognized that small sample of OA with VAD limit stratified analysis for other covariables (LINE 380-381)
In the discussion, we acknowledge that the association observed (of VAD) may be result of residual confounding due to inflammation despite the adjustment of the models by CRP, AGP and IL-6 (LINE 367-369). This is important because through models (1-3), we can observe how the coefficient change due to the adjustment for these biomarkers (including IL-6).
To address our research question, we assumed that retinol levels indicate the nutritional status of VA, but as we have previously recognized, retinol values may be result of inflammatory effects rather than the nutritional status of VA. Adjusting for biomarkers of inflammation is a helpful strategy for account for such effects. Nevertheless, exploring the correlation between intake of retinol equivalents in the diet and retinol serum in the OA from this study, the analysis showed a positive correlation (Rho=0.11, p<0.001); indicating that retinol values may also reflect the nutritional status of VA. This information was added to discussion (LINE 369-375).
It is important to clarify that we do not denied the contribution of other factors that explain higher hepcidin levels, as CRP, AGP or IL-6 (and iron indicators as we describe in the section of statistical analysis). However, the relevant of this study is that despite adjusting for those biomarkers, VA contribute to explain the variability of hepcidin trough higher levels, in a lower magnitude than those of AGP, CRP or IL-6.
Minor comment:
Introduction
1. Line 59-61, line 70 and line 215. This reference [19] is not published and should be deleted.
Response The reference 19 (causes of anemia in Mexican older adults) is an unpublished work and it is in process for publication. We changed the format of reference because that information is available in the doctoral thesis. Following the instruction for authors -section References, we adapted the citation in thesis format, as follow:
De la Cruz-Góngora, V; et al. “Causas de anemia en adultos mayores: papel de la hepcidina, vitamina A y vitamina D”. Tesis de Doctorado. Escuela de Salud Pública de México, Cuernavaca, Morelos; México, Agosto 27, 2018.
Methods
1. Line 67. The authors recruited 829 OA (ages ≥ 60y) to understand the causes of anemia. However, only 35.5% of participants have anemia (Table1). Criteria for selection of OA are unclear and should be clarified.
Response Corrections on sample size was done in the methods and result sections, table 1 and table 2. Criteria for selection of OA were clarified in the methods section as follows:
“For the sampling procedure, we used a stratified multistage cluster sample design. If in the household two OA were living together, only one was randomly selected to participate in the study. Selection criteria: all ambulatory OA were invited to participate in the study. Exclusion criteria applied to OA with any leg amputation, dementia (or complete dependence of their caregiver), or were using supplementary oxygen, or if for some other condition they were immobilized or resting full-time in bed. (LINE 71-76).
2. Line 77. There is a circadian rhythm in VD levels [Eur J Endocrinol. 2002,146, 635-42]. This may affect the results. Is the time of blood sampling similar in all OA?
Response Blood samples in all OA were collected with at least 8 hrs. of fasting. The time of collected blood sample was between 6-9 am, previous arranged appointment with the OA. Considering this, the time of blood sampling was similar in all OA. In the methods section, line 83, we clarify the time of blood sample collection. “Fasting venous blood samples were drawn and centrifuged in situ between 6:00-9:00 am, previous scheduled appointment.”
3. Line 80, 91-92. Data for homocysteine and EPO are not given. In addition, data for creatinine and GFR should be given, regardless of CKD. Is there any CKD patient on dialysis?
Response All OA were ambulatory and all those who decide participate were included in the study. Unfortunately, we did not asked specifically if they were recently under dialysis treatment. Information regarding eGFR, EPO and biomarkers of inflammation were added in tables 1 and 2.
4. Line 101. “mcg/dL” should be “mg/dL”.
Response We made the correction of mcg/dL and ug/dL to homologate to “mg/dL” in all the variables listed in the file (tables 1, methods section)
Results
1. Data for Hb should be given in Tables 1 and 2. Is there any correlation between Hb and ferritin, hepcidin, VA, VD, CRP, AGT, IL-6, creatinine and GFR?
Response We added the information of Hb (mean) in Table 1 and 2.
All correlation with Hb were significant and in the expected direction. The correlation between Hb and biomarkers were as follows: rho= 0.11, p=0.0016 for ferritin; rho= -0.09, p=0.01 for hepcidin; rho=0.08, p=0.0139 for retinol; rho= 0.127, p=0.0003 for vitamin D; rho= -0.072, p=0.04 for CRP; rho= -0.141, p=0.0001 for AGP; rho= -0.164, p<0.001 for IL-6; rho= -0.103, p=0.0037 for creatinine; and rho=0.279, p<0.001 for eGFR.
This information was not added in the article because it corresponds to a specific question addressed in the old reference [19] (main causes of anemia in OA); which is a manuscript in process of publication.
2. While percentage of anemia is very high (66.7%) in OA with VAD than in those without (34.7%, Table2), percentage of anemia in a whole group is 35.5% (Table 1).
Response VAD is a recognized risk factor for anemia. So, it is expected that people with VAD had higher prevalence of anemia that those without. The 66% of OA with anemia in the VAD category is an expected frequency at biological level given that VA deficiency increase the probability of having anemia. Because the number of OA with VAD was small, it did not influenced the mean of the parameter.
3. It is hard to understand the data only by frequency. The actual number of the patients in each category should be given in Tables 1 and 2. In Table 1, the number of the patients is 829 but it should be 803 as described in line 71. In Table 2, at least the number of the patients in each group, namely VD≥50nml/L, VD<50nmol/L, retinol≥20mg/dL and retinol<20mg/dL should be given.
Response We added the actual number of OA in table 1 and Table 2, excluding those who were taking VD supplements. (Table 2)
3. Table 2 only shows percentage of anemia but what type of anemia is not clarified.
Response We added the type of anemia in table 2.
4. Line 159-162. Prevalence of hypertension in OA with and without VDD is not significant (P>0.05, Table 2).
Response We made the pertinent correction in the description of results, no describing hypertension as significant variable by VDD category. (LINE 177-180)
5. Line 162-165. OA with VAD had low prevalence of VDD. Actual number of the patients with VDD (VD<50nmol/L) should be given in Table 2.
Response We added the actual number of the patients by VA and VD status in the table 2. In table 1 we added the number of OA with information collected.
6. Line 159-165. The authors stated that anemia is more prevalent in OA with VDD and VAD than those without. Table 2 shows high prevalence of VB12 deficiency that causes anemia in these groups. Does it mean that malnutrition is more prevalent in OA with VAD and VDD than those without? Or. Does it mean that prevalence of VB12 deficiency anemia is higher in OA with VAD and VDD than those without?
Response
After excluding those OA who were receiving VD supplements, the prevalence of B12 deficiency was not different by VAD nor VDD than those without. In addition, VB12 deficiency anemia was not different between VDD nor VAD categories.
This do not exclude the possibility that malnutrition in OA coexists with multiple nutritional deficiencies.
7. Results show no difference in iron deficiency and low ferritin in OA regardless of VD and VA status, whereas anemia of inflammation is only higher is OA with VAD but not OA with VDD. The authors stated that adjusting for VD supplements did not change the results [line299] but they did not examine the data when such patients receiving VD supplements are excluded.
Response We re-analyzed all data excluding those OA who were taking VD supplements. The main association did not changed nor the conclusion originally observed between VD status and hepcidin (table 3).
In addition, actual number of OA with VDD (78 patients) is higher that of those with VAD (27 patients) and thus statistical significance is stronger in the former group than in the latter group. It is not clear how lower prevalence of OA with VDD in this study than in other countries [line300-] can explain the reason why VDD is not associated with higher hepcidin in this study.
Response The study was conducted in non-institutionalized OA in a very sunny and tropical region of Mexico. Thus, we expect that they have some degree of occasional sun exposure during most of the year that may explain the lower prevalence of VDD measured by 25(OH)D. On the other hand, the lack of association of VDD and hepcidin was a finding of our study that could be related to this low prevalence of VDD, and thus most of the subjects had adequate levels of VD, limiting the possibility of observing an association. Some of the reason of lack of association between VD and hepcidin was mentioned in the discussion section (LINE 330-366).
8. Prevalence of anemia of inflammation is only higher in OA with VAD than those without. However, its prevalence is not different between OA with and without VDD. Prevalence of low serum iron, high ferritin, parameters of inflammation (CRP, IL-6 and AGP), obesity and comorbidities associated with inflammation are not different between OA with and without VDD. Only difference is found in prevalence of VB12 deficiency in OA with VDD than those without. Prevalence of anemia is higher in OA with VDD than those without. Does it mean that VB12 deficiency anemia is more prevalent in OA with VDD than those without?
Response VB12 deficiency anemia was not different between VDD nor VAD categories. The B12 deficiency indicator was constructed regardless of anemia condition.
Prevalence of B12 deficiency was not different by VAD nor VDD than those without. This do not exclude the possibility that malnutrition in OA coexists with multiple nutritional deficiencies.
9. Line 170-174. The authors stated that serum hepcidin levels were significantly lower in OA with VAD than in OA with normal retinol (Fig1a). However, this should be “serum hepcidin levels were significantly higher in OA with VAD than in OA with normal serum retinol (Fig1b).” Namely, Fig1a should be Fig1b and Fig1b be Fig1a.
Response We made the respective correction of legends in figure 1 as well in their description in the Results section. (LINE 199-203)
10. The definition of inflammation (CRP>5mg/L, AGP1g/L) according to ref 23 and 30 may be used in this study. However, CRP was divided into 3 groups in Table 2; 0-3, 3-10 and >10mg/L instead of 0--5 and >5mg/L as shown in Table1. Why IL-6 levels were divided into ≤10 and >10 pg/mL? What are the normal values of IL-6? The definition of inflammation as measured by CRP, IL-6 and AGP should be clearly described.
Response Categories of inflammation were considered using AGP and CRP combination according Turnham’s equation: reference, incubation, early convalescence and late convalescence. This equation used the cut-off of CRP>5mg/L and AGP>1g/L. Information of IL-6 was not considered in the definition.
We describe the status of inflammation in the methods section (Line 114-115,) and we present the frequency of OA according to these stages of inflammation.
There is no “conventional cut-off “ to define normal vs higher IL-6 values. We used the cutoff IL6>10 pg/mL since Den Enzel used [26], in order to compare the prevalence of this condition considering OA characteristics between both studies.
11. Percentage of OA receiving drugs is different from that of those with NSAID plus SAID. Why do OA with VAD have lower prevalence of drugs (other than NSAID or SAID) as compared to those without?
Response We do not know the reason of lower prevalence of use of medication between OA with VAD in comparison to those without. We supposed that since prevalence of comorbidities were not different according to VAD status, maybe OA with VAD could have lower access or adherence to pharmacological treatment.
12. High levels of ferritin and hepcidin and low levels of serum iron are characteristic of FIDA or anemia of inflammation. In this study, anemia, high inflammation, low serum iron and high ferritin and hepcidin, characteristic of FIDA, are associated with OA with VAD as compared to those without. Since hepcidin data are not adjusted for inflammation, this finding suggests that inflammation may induce VAD, leading to higher ferritin and hepcidin and subsequent anemia of inflammation. In fact, the levels of VA are inversely correlated with the levels of inflammatory markers (CRP, IL-6) and this negative correlation was stronger for anemics [line 179-189].
Response After excluding those OA and re-analyzing again all statistical models, all hepcidin data were adjusted by inflammatory biomarkers (AGP and CRP) (figure 1, and Table 3, model 1 and 2) as well as Il-6 (Table 3, model 3). Nevertheless, we acknowledge that despite all this adjustment, residual confounding due to inflammation may persist and VA indicator may reflect some degree of inflammation. (Discussion section, limitation of the study: line 366-374). We acknowledge that VA status may be a consequence of inflammation. To account for this, we adjusted for 3 biomarkers of inflammation. Nevertheless, FIDA may be present in malnutrition, a condition that confer a higher susceptibility of infection due to lower immunity, thus to stablish the causality between malnutrition-inflammation-malnutrition remain a challenge in this OA population.
In children [Zimmerman, et al ], supplementation of VA have shown a higher iron mobilization stores without changing the ratio of ferritin/stfr. This suggest that VA modulate (through immune system) hepcidin expression to lead iron store mobilization from ferritin (independent of their effect on EPO levels expression). Unfortunately, the evidence that show such an association between VA and hepcidin has not being well documented in humans and it is limited to rodents, to compare our results. Because lack of such evidence, we proposed this hypothesis considering all possible variables that may confound and explain the variability of hepcidin, VA and VD.
Zimmermann M.B., Biebinger R., Rohner F., Dib A., Zeder C., Hurrell R.F., Chaouki N. Vitamin A supplementation in children with poor vitamin A and iron status increases erythropoietin and haemoglobin concentrations without changing total body iron. Am. J. Clin. Nutr. 2006;84:580–586
Discussion
1. Line 210-211. If so, same conclusion should be obtained in OA without inflammation.
Response We agree with the comment of the reviewer. In effect, the analysis showing the VAD and hepcidin association should considered those without inflammation. However, if we restrict the analysis to those without inflammation or minimal inflammation we found only 7 OA with VAD, the small number of OA with VAD limits the power of the study.
We arranged the paragraph in the discussion, clarifying that such association occurred when it was adjusted by biomarkers of inflammation (line 243-244):
“In this study, we found that retinol levels were associated with hepcidin concentrations, after adjusting for the inflammatory condition, since VAD were associated to higher hepcidin levels in OA.”
In the discussion, we acknowledge that the association observed (of VAD) may be result of residual confounding due to inflammation despite the adjustment of the models by CRP, AGP and IL-6 (LINE 366-374). To address our research question, we assumed that retinol levels indicate the nutritional status of VA, but as we have previously recognized, retinol values may be result of inflammatory effects rather than the nutritional status of VA. Adjusting for biomarkers of inflammation is a helpful strategy for account for such effects. Nevertheless, exploring the correlation between intake of retinol equivalents in the diet and retinol serum in the OA from this study, the analysis showed a positive correlation (Rho=0.11, p<0.001); indicating that retinol values may also reflect the nutritional status of VA.
2. Throughout discussion the authors refer to anemia but what type of anemia should be clarified.
Response In the discussion we refer the type of anemia in the OA participants in our study (LINE 248-251). In those cases where we refer to other studies, the type of anemia is not describe by the authors.
3. Line 215-220. As described, ref19 should be deleted from the manuscript.
Response The reference 19 (causes of anemia in Mexican older adults) is an unpublished work and it is in process for publication. We changed the format of reference because that information is available in the doctoral thesis. Following the instruction for authors -section References, we adapted the citation in thesis format, as follow:
De la Cruz-Góngora, V; et al. “Causas de anemia en adultos mayores: papel de la hepcidina, vitamina A y vitamina D”. Tesis de Doctorado. Escuela de Salud Pública de México, Cuernavaca, Morelos; México, Agosto 27, 2018.
Line287. In this study, the authors stated that the effect of inflammation was adjusted for evaluation of hepcidin. However, they discussed that OA were associated with inflammatory condition possibly different from other study [ref40].
Response OA participants in this study came from the Southern Region of Mexico -a less developed region- where characteristic are no similar to those OA from developed countries nor from the national older Mexican adult population. This population studied, came from the Maya region, had higher prevalences of anemia and comorbidities than those reported at national level, as well as higher prevalence of frailty, AIVD and AVD. All these conditions make us think that this population might have a different inflammatory profile than other OA populations and other inflammatory conditions (i.e. inflammatory bowel disease, autoimmune disease, among others).
Recognizing all these factors, will help us to understand their contribution to anemia (particularly, AI), which actually is a serious and rising health problem in OA Mexico.
Figures
1. The name of variable should be indicated on vertical line.
Response Name of variables of the figures were indicated on vertical line
Tables
1. Table 1 shows 829 OA but serum and hematological data are available in 803 OA (line 71).
Response We made the respective correction of the number of OA analyzed, restricted from 803 with serum samples to 783 OA, because we exclude those who were taking supplements of VD (n=20) as suggested.
2. Tables 1 and 2 should show raw data for hepcidin. CRP, IL-6, AGT, ferritin, EPO, serum albumin, leptin, total cholesterol, creatinine and GFR as well as the number of patients with CRP≥5mg/L and <5mg and those with and without high IL-6.
Response Information on CRP, IL-6, AGT, ferritin, EPO, creatinine and eGFR, CRP>5mg/L and Il>10 was added in table 1 and 2.
3. Why are p values missing in some variables of Table 2?
Response We reported a global p values associated to the Fisher’s test (this test analyzes the statistical independence of both categorical variables) when analyzing by VA/VA status; so, there are no missing p values because the reported values apply for all categories of comparison.
4. The unit of VD in first raw of Table 2 should be “nmol/L”.
Response We made the correction of “nmol/L” in table 2.
Miscellaneous
The authors should be careful about the use of abbreviations and the order of their appearance in the text.
Response Thank you, we corrected and described the abbreviations used in the order of appearance.

Reviewer 2 Report
The study seems to be fairly limited. The authors observed that serum hepcidin correlates with retinol concentration. Interestingly serum retinol correlates with serum iron. The paper is nicely written however too little attention is placed on effects of iron on hepcidin. In the discussion, section authors mention that iron overload can influence blood hepcidin but what about the signaling role of blood iron and role of transferrin receptor 2. One can expect that in persons with higher serum iron the hepcidin concentration will be higher.
The authors dedicated considerable space of their manuscript talking about regulation of hepcidin biosynthesis and I consider it as very valuable, however, if analysis of the relationship between serum iron and hepcidin will be evaluated significant changes in the manuscript are expected.
Author Response
Reviewer 2
The study seems to be fairly limited. The authors observed that serum hepcidin correlates with retinol concentration. Interestingly serum retinol correlates with serum iron. The paper is nicely written however too little attention is placed on effects of iron on hepcidin. In the discussion, section authors mention that iron overload can influence blood hepcidin but what about the signaling role of blood iron and role of transferrin receptor 2. One can expect that in persons with higher serum iron the hepcidin concentration will be higher.
Response: We agree with the comment of the reviewer. Iron and hepcidin was not the main association analyzed in this study, taking into account that hepcidin has been considered the principal hormone involved in iron regulation. Hepcidin expression has positive regulators (inflammation, iron overload) and negative regulators (hypoxia and anemia/erythropoiesis). Acknowledging these factors that may explain hepcidin variability, we focus our analysis in the positive regulators pathways, that may counteract the hepcidin expression through modulating the inflammatory profile by VD and VA, two potent anti-inflammatory vitamins.
Because aging process confer an immunedysregulation through a proinflammatory profile, we address our research question considering that VD and VA can act as anti-inflammatory agents related to hepcidin expression. Nevertheless, we acknowledge that blood iron is an important stimulus that may explain higher hepcidin levels. For that reason, such variables were considered in the adjusted models to control the confounding in the association between VD and VA status with hepcidin.
The authors dedicated considerable space of their manuscript talking about regulation of hepcidin biosynthesis and I consider it as very valuable, however, if analysis of the relationship between serum iron and hepcidin will be evaluated significant changes in the manuscript are expected.
Response: The focus of the biosynthesis of hepcidin was centered in the inflammation pathway stimulus rather than iron stimulus status because VA and VD indirectly modulate iron reserves through immunomodulation (additional to the EPO expression by VA). This is the reason why we focus to address the specific question of VA and VD considering in the regression models the iron reserves (ferritin) as well as transferrin receptor.
The evidence of VA on anemia has been consistent in the literature. The meta-analysis reported by Da Cunha et al, showed that VA-Sup on anemia and iron status in different life stages (not including OA) reduced the risk of anemia by 26% by improving hemoglobin and ferritin levels in individuals with low serum retinol (but without effects on the prevalence of ID by using serum ferritin as biomarker). Zimmerman, et al, showed in children that supplementation of VA was associated to higher iron mobilization stores without changing the ratio of ferritin/stfr. However, the question that remains unanswered is whether VA might modulate (through immune system) the hepcidin expression to lead iron store mobilization from ferritin (independent of their effect on EPO levels expression). Unfortunately, the evidence that show such an association between VA and hepcidin has not been well documented in humans and it is limited to rodents.
Because lack of such evidence, we proposed this hypothesis considering all possible variables that may confound and explain the variability of hepcidin in association with VA and VD (i.e biomarkers of inflammation, iron status, frailty, among others).
da Cunha, M. de S. B.; Campos Hankins, N. A.; Arruda, S. F. Effect of vitamin A supplementation on iron status in humans: a systematic review and meta-analysis. Crit. Rev. Food Sci. Nutr. 2018, 00–00, doi:10.1080/10408398.2018.1427552.
Zimmermann M.B., Biebinger R., Rohner F., Dib A., Zeder C., Hurrell R.F., Chaouki N. Vitamin A supplementation in children with poor vitamin A and iron status increases erythropoietin and haemoglobin concentrations without changing total body iron. Am. J. Clin. Nutr. 2006;84:580–586

Round 2
Reviewer 1 Report
I have carefully read the revised manuscript. All issues raised are adequately addressed. However, some minor points remain to be addressed as described below.
1. Line132. “exclude” should be “excluded”.
2. Fig 1 with a vertical line may be better.
3. Line 264. “…by inflammation. [19]” should be corrected.
4. Line 304. The paragraph “Contrary to…” should start a new line.
5. Line 303 and 304. “chronic inflammation.[40]” and “adverse events. [41,42]” should be “chronic inflammation [40].” and “adverse events [41,42].”.
Author Response
Reviewer 1
1. Line132. “exclude” should be “excluded”.
Answer: We made the correction of the word. Line 132.
2. Fig 1 with a vertical line may be better.
Answer: In Figure 1, we added the vertical line in Y-axis.
3. Line 264. “…by inflammation. [19]” should be corrected.
Answer: Correction has been made.
Line 304. The paragraph “Contrary to…” should start a new line.
Answer: Correction has been made.
5. Line 303 and 304. “chronic inflammation.[40]” and “adverse events. [41,42]” should be “chronic inflammation [40].” and “adverse events [41,42].”.
Answer: The punctuation mark has been corrected and placed at the end of the reference (or sentence), through all the section (line 246, 266, 303 and 304). Thank you.

Reviewer 2 Report
no more comments
Author Response
There were no comments